# Assessment of Microcirculatory Dysfunction by Measuring Subcutaneous Tissue Oxygen Saturation Using Near-Infrared Spectroscopy in Patients with Circulatory Failure

**DOI:** 10.3390/diagnostics14212428

**Published:** 2024-10-30

**Authors:** Jun Sato, Atsushi Sakurai, Shingo Ihara, Katsuhiro Nakagawa, Nobutaka Chiba, Takeshi Saito, Kosaku Kinoshita

**Affiliations:** Department of Acute Medicine, Division of Emergency and Critical Care Medicine, Nihon University School of Medicine, 30-1 Oyaguchi Kamimachi, Itabashi-ku, Tokyo 173-8610, Japan; sato.jun@nihon-u.ac.jp (J.S.); ihara.shingo@nihon-u.ac.jp (S.I.); nakagawa.katsuhiro@nihon-u.ac.jp (K.N.); chiba.nobutaka@nihon-u.ac.jp (N.C.); saitou.takeshi@nihon-u.ac.jp (T.S.); kinoshita.kosaku@nihon-u.ac.jp (K.K.)

**Keywords:** microcirculation, subcutaneous tissue, spectroscopy, near-infrared

## Abstract

Background: Patients with circulatory failure have high mortality rates and require prompt assessment of microcirculation. Despite the improvement in hemodynamic parameters, microcirculatory dysfunction persists. We measured subcutaneous regional tissue oxygen saturation (rSO_2_) with near-infrared spectroscopy (NIRS), which can assess microcirculation in patients with circulatory failure. Methods: A finger-worn oximeter with NIRS measured rSO_2_ in the forehead, thenar eminence, thumb, and knees. First, the rSO_2_ was measured in healthy adult volunteers (*n* = 10). Circulatory failure was defined as a systolic blood pressure ≤ 90 mmHg and lactate ≥ 2 mmol/L. The study included 35 patients without circulatory failure and SOFA score of 0 at ICU admission and 38 patients with circulatory failure at ICU admission. Both groups included a single-center prospective study of patients who were transported to the ICU of the Nihon University Hospital. The rSO_2_ was measured only on ICU admission in the non-circulatory failure group and later in the circulatory failure group. Results: In the volunteer group, rSO_2_ at each site was approximately 58%. The rSO_2_ was significantly lower in the circulatory failure group than in the non-circulatory failure group (knee, *p* < 0.01). In the circulatory failure group, knee rSO_2_ showed a significant negative correlation with SOFA score (Day 0, ρ = −0.37, *p* = 0.02; Day 1, ρ = −0.53, *p* < 0.01; Day 2, ρ = −0.60, *p* < 0.01). Conclusions: Subcutaneous knee rSO_2_ was associated with SOFA score and was considered an indicator of microcirculatory dysfunction and organ damage.

## 1. Introduction

Patients with circulatory failure have a high mortality rate and require treatment based on rapid assessment of microcirculation [1,2]. To date, such patients have been treated with the goal of normalizing systemic hemodynamic parameters such as blood pressure, central venous pressure, urine output, and central venous blood oxygen saturation [3]. However, microcirculatory dysfunction persists even after the improvement in these indices [4], suggesting a loss of hemodynamic coherence between macrocirculation and microcirculation [5]. In 2018, a consensus was announced by the task force of the European Society of Intensive Care Medicine regarding the evaluation of sublingual microcirculation, and the importance of evaluating microcirculation in patients with circulatory failure was pointed out [6]. The methods for assessing microcirculation include laser doppler techniques, video microscopy, and near-infrared spectroscopy (NIRS) [7]. There have been many studies on the evaluation of sublingual microcirculation using handheld video microscopes; however, the measurement accuracy is affected by saliva bubbles [8]. There are only a small number of cases using the laser doppler technique; therefore, it appears that it is not used widely in clinics [9]. In the evaluation of microcirculation using NIRS, the measurement of regional tissue oxygenation saturation (rSO_2_) in muscles is the main focus due to the influence of the measurement depth of NIRS; however, it is not being used as a sufficient indicator of microcirculation [10]. The skin is sensitive to circulatory failure because mottling signs appear early in patients with circulatory failure [11]. Therefore, we considered it worthwhile to examine the usefulness of microcirculation assessments by measuring the subcutaneous rSO_2_. In patients with circulatory failure, we measured rSO_2_ 2–5 mm subcutaneously using a finger-mounted tissue oximeter with the NIRS technique [12] (Toccare Astem Co., Ltd., Kawasaki, Japan) to investigate its relationship with microcirculation. Toccare uses NIRS technology to detect the differences in the absorbance of oxygenated (O_2_Hb) and deoxygenated (HHb) hemoglobin to calculate the concentrations of O_2_Hb and HHb. Thus, rSO_2_ = {[O_2_Hb]/([O_2_Hb] + [HHb])} × 100 (%) was calculated [12]. A low rSO_2_ measured by Toccare indicates low O_2_Hb in the subcutaneous microcirculation, which indicates microcirculatory dysfunction.

There are few reports on subcutaneous tissue rSO_2_ measurements and the use of Toccare in adults [13,14,15,16]. We first measured rSO_2_ in healthy adult volunteers using subcutaneous tissue. Subsequently, we measured rSO_2_ in multiple subcutaneous tissue locations using Toccare in patients with non-circulatory and circulatory failure. This study aimed to compare the differences in rSO_2_ between patients with and without circulatory failure and analyze the relationship between rSO_2_ and macrocirculation, microcirculation, and organ dysfunction.

## 2. Materials and Methods

### 2.1. Study Population

This study included three groups: volunteer, non-circulatory failure, and circulatory failure groups. The volunteer study was for healthy adults. When the volunteer study was approved by the Ethics Committee of Nihon University Hospital in 2020, the legal age of adulthood in Japan was 20 years, and the patients of this study were determined to be 20 years of age or older. As we were unable to recruit any underage volunteers, only adults were included in this study. The inclusion criteria for the non-circulatory failure group were as follows: patients who were admitted to the intensive care unit (ICU) of Nihon University Hospital between 1 April 2023 and 31 March 2024, with systolic blood pressure (sBP) > 90 mmHg, lactate level < 2 mmol/L, and sequential organ failure assessment (SOFA) score of 0. The exclusion criteria were patients under 20 years of age and those who were unable to consent to this study. The ICU at Nihon University Hospital also accepted patients who were judged to be in serious condition but did not have any abnormalities in their vital signs; therefore, this group was also included in the study. The inclusion criteria for the circulatory failure group were patients who were admitted to the ICU of Nihon University Hospital between 15 June 2020 and 31 March 2022 and had sBP ≤ 90 mmHg or lactate level ≥ 2 mmol/L. The exclusion criteria were patients who had received cardiopulmonary resuscitation, those under 20 years of age, and those who were unable to consent to this study. The non-circulatory and circulatory failure groups were part of a single-center prospective study at the Nihon University Hospital. In both groups, consent was obtained primarily from the patient; however, in cases where this was not possible, consent was obtained from the next of kin. There is normal data for fetuses but not for children in studies using Toccare [12]. This volunteer study targeted patients aged 20 years and above. Since the normal rSO_2_ in children measured with Toccare remains unknown, the non-circulatory and circulatory failure groups targeted patients aged 20 years and above. The volunteer (20200205), non-circulatory failure (20230305), and circulatory failure (20200602) studies were approved by the Ethics Committee of Nihon University Hospital.

### 2.2. Measurement Site and Method for rSO_2_

In the three studies, we used Toccare to measure rSO_2_ at the forehead, palmar aspects of the thenar eminence and thumb, and extensor aspect of the knee joint. As the appropriate subcutaneous rSO_2_ measurement site for microcirculation evaluation has not been identified, we chose these measurement sites as part of an exploratory study. The reason for selecting these four sites was that the knee and thenar eminence were used in past NIRS studies [10,17], and the forehead was chosen to compare with the extremities. In addition, the thumb is slightly more proximal than the thenar eminence, and we chose it to investigate whether there is a difference in rSO_2_ between the proximal and distal sides of the extremities. These four sites were selected on either the left or right side of each patient or volunteer, and all four sites were measured on the same side. The rSO_2_ varies depending on tissue scattering and absorption coefficients [18]. After confirming that there were no visible blood vessels near the measurement site, the Toccare probe was held in contact with the skin for 10 s, and the value was recorded when it stabilized.

### 2.3. Measured Data

The volunteer group was assessed for rSO_2_ at the above four measurement sites, blood pressure, pulse rate, and Glasgow Coma Scale (GCS) in sitting and resting positions after obtaining written consent. In the non-circulatory failure group, the following parameters were measured on admission at the ICU: rSO_2_ at the above four measurement sites, blood pressure, pulse rate, GCS, white blood cell (WBC), hemoglobin (Hb), hematocrit (Hct), platelet (Plt), prothrombin time (PT), activated partial thromboplastin time (APTT), D-dimer, total-bilirubin (T-bil), blood urea nitrogen (BUN), creatinine (Cre), sodium (Na), potassium (K), C-reactive protein (CPR), blood glucose, and arterial blood gas analysis was performed. In the circulatory failure group, the following parameters were measured at 0, 6, 12, 24, and 48 h after admission to the ICU: rSO_2_ at the above four measurement sites, blood pressure, pulse rate, GCS, blood counts, and arterial blood gas analysis was performed. In addition, the following blood tests were performed on admission at the ICU and on the first and second days of hospitalization: WBC, Hb, Hct, Plt, PT, APTT, D-dimer, T-bil, BUN, Cre, Na, K, CPR, blood glucose, and arterial blood gas analysis. The causes of circulatory failure were classified as hypovolemic, distributive (further classified as septic or anaphylactic shock), cardiogenic, or obstructive. The attending physician provided treatment deemed optimal for a patient with circulatory failure.

### 2.4. Statistical Analyses

Data analysis was performed using the Statistical Package for the Social Sciences (SPSS Inc., Version 16; Chicago, IL, USA). Normality tests were performed using the Shapiro–Wilk test. For non-parametric continuous variables, medians and interquartile ranges (IQR) were presented; two-group tests were performed using the Mann–Whitney U test, and tests for three or more groups required the Kruskal–Wallis and Steel–Dwass methods, which could not be performed in SPSS and were therefore performed in Easy R (EZR) [19]. Categorical variables are expressed as numbers (*n*) and percentages (%), and Fisher’s exact test was used to determine significant differences between the groups. The correlation coefficient (ρ) and *p*-value were calculated to determine the relationship between the two non-parametric continuous variables using the Spearman rank correlation coefficient. Statistical significance was set at *p* < 0.05. To evaluate whether rSO_2_ at ICU admission can predict the occurrence of circulatory failure (sBP ≤ 90 mmHg or lactate level ≥ 2 mmol/L was defined as circulatory failure), receiver operating characteristic (ROC) curves were created using rSO_2_ in the non-circulatory and circulatory failure groups at ICU admission. The optimal cut-off value was determined based on the maximum value of the sum of sensitivity and specificity, which represents the minimum distance from the top left corner to a point on the ROC curve.

To determine the factors affecting rSO_2_ in the circulatory failure group, we performed multiple linear regression analysis with the forced entry method to examine the relationship between rSO_2_ and age, Hb level, and SOFA score. Age and Hb were selected as independent variables for multiple linear regression analysis because the rSO_2_ measured by Toccare was reported to be negatively correlated with age, and the absorbance of Hb was the principle of rSO_2_ measurement [12,16]. We also selected the SOFA score as an independent variable because microcirculatory failure is reported to be associated with organ damage [20]. Blood pressure was not included as an independent variable because the SOFA score includes blood pressure as an assessment of circulation. The variance inflation factor (VIF) was used to evaluate the multicollinearity among independent variables; a VIF > 10 was considered multicollinear. Residual analysis was performed to assess the appropriateness of multiple linear regression equations. The normality of the residuals was analyzed using the Shapiro–Wilk test.

## 3. Results

### 3.1. Composition of Each Group

The volunteer group consisted of 10 healthy adult volunteers (five males and five females, aged 20 years or above, with no medical history and no oral medications) who provided consent to participate in the study. In the non-circulatory failure group, 122 patients fulfilled the selection criteria. Of these, 48 patients under the age of 20 years were excluded, and 39 other cases from whom consent could not be obtained were also excluded, leaving a final analysis target of 35 patients. In the circulatory failure group, 259 patients fulfilled the selection criteria. Of these, 129 cases of cardiac arrest were excluded, and 92 other cases from whom consent could not be obtained were also excluded, leaving a final analysis target of 38 patients. The demographic and clinical characteristics of the volunteers, non-circulatory failure group (at ICU admission), and circulatory failure group (at ICU admission) are shown in Table 1.

The median ages of the volunteer, non-circulatory, and circulatory failure groups were 34.0 (IQR: 31.0–40.0), 62 (49.0–78.5), and 73 (60.0–84.0) years, respectively. There was no significant difference in age between the non-circulatory and circulatory failure groups (*p* = 0.06). The reasons for ICU admission in the non-circulatory failure group were trauma (*n* = 15, 42.9%), cardiovascular disease (*n* = 14, 40%), other internal medical diseases (*n* = 4, 11.4%), and overdose (*n* = 2, 5.7%). The category of “trauma” included cervical spinal cord injuries and fractures of the extremities, while the category of “cardiovascular disease” included pleurisy and myocardial infarction. The causes of shock in the circulatory failure group were septic (*n* = 18; 47%), hypovolemic (*n* = 13; 34%), cardiogenic (*n* = 3; 8%), anaphylactic (*n* = 2; 5%), and obstructive (*n* = 2; 5%) shocks. The sBP, lactate, and hemoglobin (Hb) levels of the circulatory failure group after admission to the ICU are shown in Appendix A. After admission to the ICU, the sBP tended to increase, and lactate and Hb levels tended to decrease.

### 3.2. rSO_2_ in Each Group

Figure 1 shows the rSO_2_ values for each site in the volunteer, non-circulatory, and circulatory failure groups. The rSO_2_ values for the circulatory failure group in Figure 1 were measured 0, 6, 12, 24, and 48 h after ICU admission.

In the volunteer group, the median rSO_2_, including four measurement sites, was 58% (IQR: 57–60%). Using the Kruskal–Wallis method, the difference between the median rSO_2_ for the four sites measured in the volunteer group was *p* = 0.026; however, when the median rSO_2_ for all sites was compared between the two groups using the Steel–Dwass method, the *p*-value was >0.05, for all combinations. In other words, the rSO_2_ in the volunteer group did not differ according to the site of measurement. In the non-circulatory and circulatory failure groups at ICU admission, the median rSO_2_, including four measurement sites, was 54% (IQR: 51–57%) and 49% (IQR: 45–54%), respectively. The rSO_2_ at all sites at ICU admission was significantly lower in the circulatory failure group than in the non-circulatory failure group. The rSO_2_ of each part of the circulatory failure group showed only slight changes over time after ICU admission.

#### Relationship Between rSO_2_ and Each Parameter

Figure 2 shows the relationship between rSO_2_, age, and Hb level for each site in the non-circulatory and circulatory failure groups at admission.

The rSO_2_ showed a significant positive correlation with age, except in the knee (non-circulatory and circulatory failure groups) and forehead (circulatory failure group). The rSO_2_ showed a significant negative correlation with Hb in all the groups except for the knee in the circulatory failure group. Appendix A shows the relationships between rSO_2_, sBP, lactate, and Hb levels in the circulatory failure group 0–48 h after ICU admission. The rSO_2_ at each site showed no correlation with sBP; however, it tended to correlate with lactate and Hb levels.

Figure 3 shows the relationship between rSO_2_, SOFA, and APACHE II scores in the circulatory failure group.

With the date of ICU admission as Day 0, rSO_2_ and SOFA scores showed significant negative correlations with the thenar eminence (Days 0 and 1), thumb (Day 0), and knee (Days 0, 1, and 2). The rSO_2_ and APACHE II scores were significantly negatively correlated at all the sites. The evaluation parameters of SOFA scores are shown in Appendix A. The rSO_2_ was positively correlated with the Glasgow Coma Scale (GCS) and negatively correlated with Cre.

Figure 4 shows the ROC curve for predicting the occurrence of circulatory failure based on rSO_2_ at ICU admission. The rSO_2_ at all measurement sites at ICU admission can significantly predict the occurrence of circulatory failure at a cut-off value of around 50–55.

Table 2 shows the results of multiple linear regression analysis using the forced entry method for rSO_2_ in the circulatory failure group. The VIFs of all the independent variables were less than 10, and there was no multicollinearity among them. All residuals of the linear multiple regression model for knee rSO_2_ were normally distributed, indicating the appropriateness of the model. The SOFA scores on Days 1 and 2 of hospitalization in the circulatory failure group explained 34% and 48% of the variance in rSO_2_, respectively, independent of age and Hb level. The results of multiple linear regression analysis for the other sites are shown in Appendix A. The rSO_2_ at the forehead (Day 1), thenar eminence (Day 0), and thumb (Day 0) were significantly associated with the SOFA score.

## 4. Discussion

To the best of our knowledge, this is the first study to examine subcutaneous tissue rSO_2_ in patients with circulatory failure in relation to severity and organ damage. First, the subcutaneous rSO_2_ in adults was found to be approximately 58% in the volunteer group, and rSO_2_ and age were significantly negatively correlated in the non-circulatory and circulatory failure groups. In a previous study using Toccare to measure buccal subcutaneous rSO_2_ in healthy Japanese women [16], the rSO_2_ ranged from 52% to 66% and was negatively correlated with age, suggesting that the rSO_2_ in our study was reasonable.

In this study, rSO_2_ was found to correlate with age and Hb level with and without circulatory failure; however, the multiple linear regression analysis showed that knee rSO_2_ in the circulatory failure group was influenced by the SOFA score, independent of age and Hb level. The results of the ROC curve also suggested that it may be possible to determine whether circulatory failure occurred based on the rSO_2_ at the time of ICU admission. Furthermore, the rSO_2_ in this circulatory failure group was negatively correlated with the SOFA and APACHE II scores; thus, subcutaneous rSO_2_ may be used to assess the severity of organ dysfunction. Additionally, among the organ failures constituting the SOFA score, impaired consciousness and renal failure were correlated with rSO_2_. This reflects the fact that the clinical findings of tissue hypoperfusion are more noticeable in the skin, kidneys, and central nervous system [2].

There were no significant differences in age or Hb levels between the non-circulatory and circulatory failure groups; however, the rSO_2_ in the circulatory failure group was significantly lower. This may also be a result of rSO_2_, which reflects microcirculatory dysfunction, leading to organ dysfunction. A previous study also reported that the muscle rSO_2_ recovery gradient at the thenar eminence reflects the autoregulatory capacity of microcirculatory dysfunction and is associated with organ dysfunction and mortality in patients with sepsis [10]. This finding also supports the validity of the present results associated with rSO_2_ and organ dysfunction.

In this study, knee rSO_2_ was correlated with SOFA scores at all time points, unlike at other sites. This suggests that the knee rSO_2_ may reflect the worsening of a patient’s condition more than other parts of the body. The knee is prone to mottling signs [21], and the skin temperatures at mottling sign sites have been reported to be decreased [11]. It has been reported that the skin temperature is regulated differently in the knee than in other parts of the body [22], with the knee having a weaker capacity for autoregulation of blood flow in response to blood pressure fluctuations and a tendency to lower skin temperature [23]. Therefore, subcutaneous rSO_2_ measurements in the knee, where blood flow autoregulation is weak, are useful for microcirculatory assessments.

In the circulatory failure group, there was no significant correlation between rSO_2_ and sBP, and although sBP improved during hospitalization, the rSO_2_ only improved slightly. This is thought to be due to a mismatch between macrocirculation and microcirculation [5]. As a result of these mismatches, the management of patients with circulatory failure may involve not only blood pressure monitoring but also the assessment of microcirculatory dysfunction with Toccare to determine the treatment strategy.

Thus, the subcutaneous rSO_2_ reflects microcirculation and is negatively correlated with organ dysfunction and disease severity. In the management of various circulation failures, measuring rSO_2_ at various sites (especially the knee) using Toccare has the potential to detect the presence of microcirculatory dysfunction.

This study had some limitations. Statistical analyses were performed without classifying circulatory failure by cause, and the differences in the cause of circulatory failure were not considered. NIRS is sensitive to variables such as skin pigmentation, tissue thickness, and ambient light; however, these factors were not adjusted in this study. This was a single-center study and involved a small number of cases; therefore, future studies involving larger and more diverse patient groups are needed. In this study, we did not compare Toccare with other methods for assessing microcirculation. Further research is needed to determine whether Toccare is superior to other methods for assessing microcirculation.

## 5. Conclusions

The subcutaneous rSO_2_ in healthy adults was found to be approximately 58%. In patients with circulatory failure, the rSO_2_ measured with Toccare showed no correlation with blood pressure but a significant negative correlation with organ damage and severity. It is possible that rSO_2_ measurements in the knees are particularly useful. In addition, it may be effective for evaluating microcirculatory dysfunction that is mismatched with blood pressure and for determining treatment strategies for circulatory failure. Further studies are needed to determine the differences in microcirculatory dysfunction according to the cause of circulatory dysfunction.

## Figures and Tables

**Figure 1 diagnostics-14-02428-f001:**
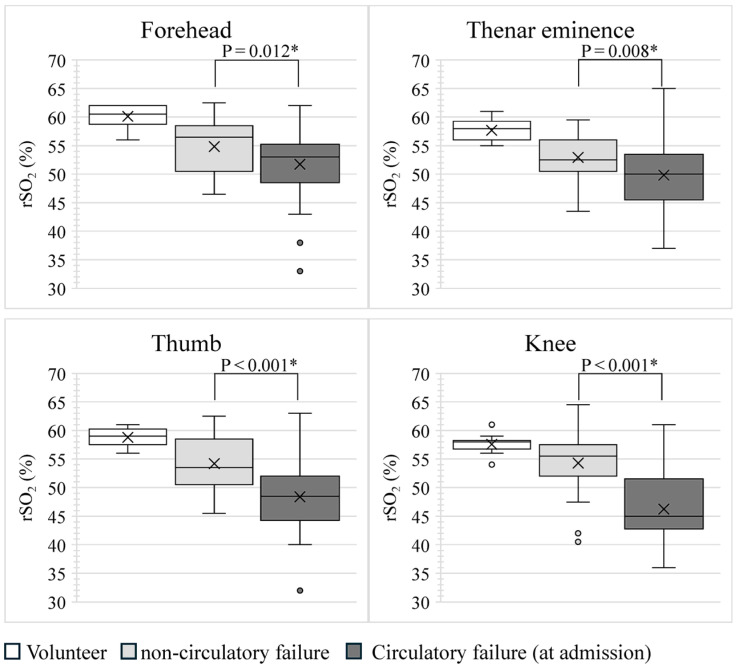
Comparison of rSO_2_ at each site. *p*-values were calculated by comparing rSO_2_ between the non-circulatory and circulatory failure groups at ICU admission using the Mann–Whitney U test. * *p* < 0.05, indicating a significant correlation. The rSO_2_ in the circulatory failure group at ICU admission was significantly lower than that in the non-circulatory failure group at all sites. rSO_2_, regional tissue oxygen saturation.

**Figure 2 diagnostics-14-02428-f002:**
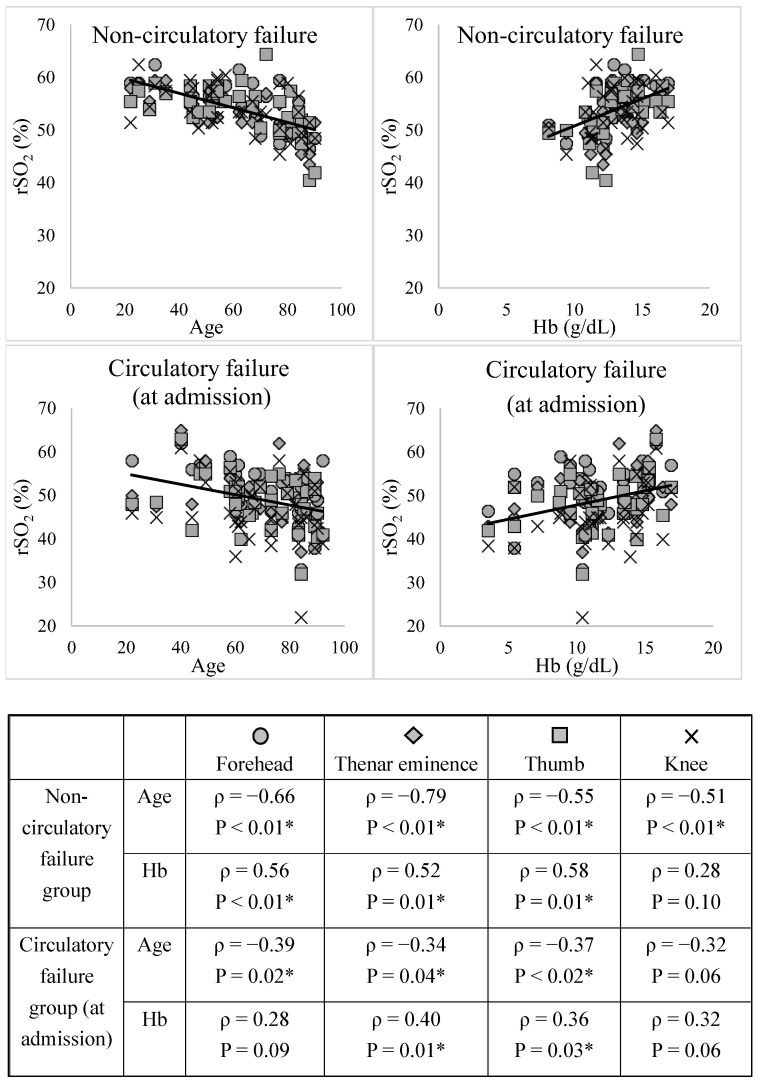
Relationships between rSO_2_, age, and Hb level at each site. The correlation coefficient (ρ) and *p*-value were calculated for the relationship between the two variables using the Spearman rank correlation coefficient. * *p* < 0.05, indicating a significant correlation. The rSO_2_ showed a significant positive correlation with age, except in the knee (non-circulatory and circulatory failure groups) and forehead (circulatory failure group). The rSO_2_ showed a significant negative correlation with Hb in all the groups except for the knee in the circulatory failure group. rSO_2_, regional tissue oxygenation saturation; Hb, hemoglobin.

**Figure 3 diagnostics-14-02428-f003:**
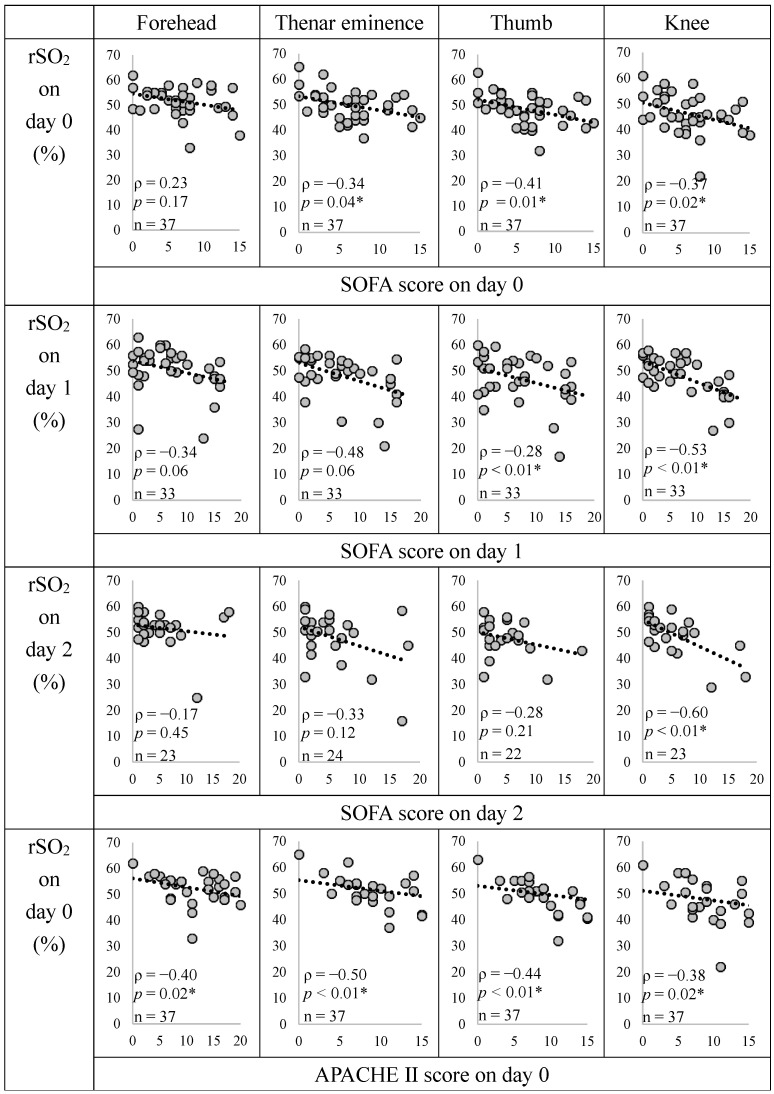
Relationship between rSO_2_, SOFA, and APACHE II scores in the circulatory failure group. The correlation coefficient (ρ) and *p*-value were calculated for the relationship between the two variables using the Spearman rank correlation coefficient. * *p* < 0.05, indicating a significant correlation. We did not create a graph for the non-circulatory failure group because the SOFA score was 0. The rSO_2_ at all measurement sites showed a significant negative correlation with the APACHE II score, and the rSO_2_ at the knee showed a significant negative correlation with the SOFA score at all time points. SOFA, Sequential Organ Failure Assessment; APACHE II, Acute Physiology and Chronic Health Evaluation II; rSO_2_, regional tissue oxygenation saturation.

**Figure 4 diagnostics-14-02428-f004:**
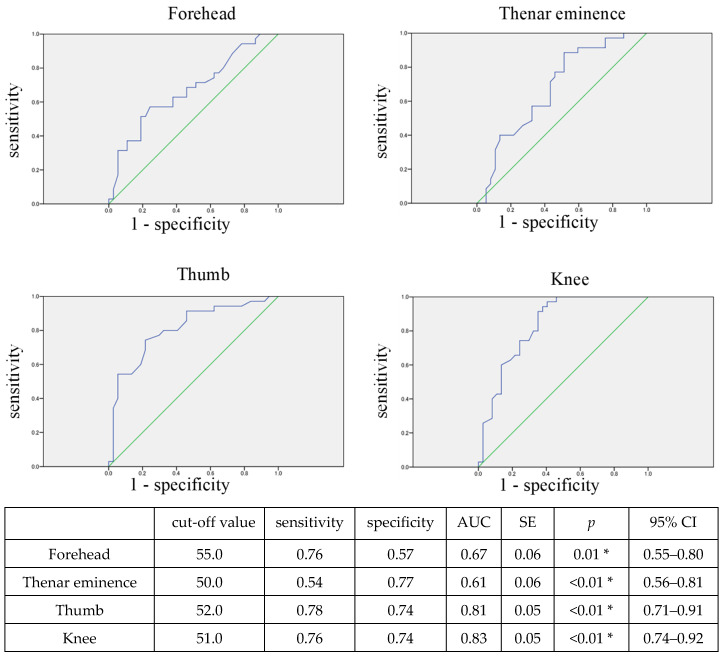
ROC curve for predicting circulatory failure using rSO_2_ at ICU admission. The blue line shows the ROC curve, and the green line shows the reference line with AUC = 0.5. The optimal cut-off value was determined based on the maximum value of the sum of sensitivity and specificity, which represents the minimum distance from the top left corner to a point on the ROC curve. * *p* < 0.05, indicating a significant correlation. The rSO_2_ at all measurement sites at ICU admission can significantly predict the occurrence of circulatory failure. rSO_2_, regional tissue oxygenation saturation; AUC, area under the curve; SE, standard error; CI, confidence interval.

**Table 1 diagnostics-14-02428-t001:** Demographic and clinical characteristics of each group of patients.

	Volunteer (*n* = 10)	Non-Circulatory Failure (*n* = 35)	Circulatory Failure (*n* = 38)	*p*-Values
Age	34 (31–40)	62.0 (49.0–78.5)	73.0 (60.0–84.0)	0.06
Male, *n* (%)	5 (50)	19 (54.3)	23 (61.0)	0.64
BMI, kg/m^2^	21.0 (20.0–22.0)	21.6 (20.1–27.1)	22.0 (18.0–25.0)	0.33
Systolic blood pressure, mmHg	111 (108–114)	133.0 (120.5–151.5)	103.0 (86.5–124.8)	<0.01 *
Diastolic blood pressure, mmHg	72 (67–76)	84 (75–97)	72 (60–79)	<0.01 *
Mean blood pressure, mmHg	85 (81–87)	101 (90–116)	82 (70–96)	<0.01 *
Heart rate, /min	68 (63–81)	75.0 (70.5–80.0)	97.0 (77.8–117.3)	<0.01 *
GCS	15 (15–15)	15.0 (15.0–15.0)	15.0 (10.3–15.0)	<0.01 *
WBC, ×10^3^/μL	N/A	8.3 (6.4–10.2)	10.6 (6.4–13.3)	0.14
Hb, g/dL	N/A	13.5 (11.9–14.6)	11.1 (9.0–12.8)	0.10
Hct, %	N/A	39.9 (35.4–42.9)	33.4 (26.2–38.8)	0.28
Plt, ×10^9^/L	N/A	226 (199.5–260.0)	132.5 (48.5–206.5)	0.01 *
PT, %	N/A	111.0 (102.0–123.5)	82.0 (59.0–130.0)	<0.01 *
APTT, s	N/A	28.0 (25.9–30.4)	30.9 (26.6–36.1)	0.66
D-dimer, μ/mL	N/A	1.1 (0.5–9.2)	5.5 (2.7–22.0)	<0.01 *
T-Bil, mg/dL	N/A	0.5 (0.4–0.7)	0.8 (0.6–1.6)	<0.01 *
BUN, mg/dL	N/A	14.2 (10.0–18.8)	27.9 (20.6–64.4)	<0.01 *
Cre, mg/dL	N/A	0.7 (0.6–0.9)	1.3 (1.1–3.1)	<0.01 *
Na, mmoL/L	N/A	139.0 (138.5–140.0)	140.0 (134.0–144.8)	0.64
K, mmoL/L	N/A	3.9 (3.8–4.3)	4.2 (3.7–4.6)	0.21
CRP, mg/dL	N/A	0.11 (0.03–0.25)	3.0 (0.2–21.1)	<0.01 *
Blood glucose, mg/dL	N/A	110.0 (100.5–129.0)	151.0 (131.0–220.0)	<0.01 *
pH	N/A	7.41 (7.39–7.44)	7.40 (7.30–7.45)	0.31
PaCO_2_, mmHg	N/A	39.2 (36.1–43.6)	31.1 (27.2–37.1)	<0.01 *
HCO_3_^−^, mEq/L	N/A	24.7 (23.8–26.4)	19.6 (16.4–22.2)	<0.01 *
Lactate, mmoL/L	N/A	1.5 (1.3–1.9)	3.7 (2.8–6.7)	<0.01 *
APACHE II score	N/A	4 (1–6)	12.0 (7.3–17.0)	<0.01 *
SOFA score	N/A	0 (0–0)	6.0 (3.0–8.0)	<0.01 *

BMI, body mass index; GCS, Glasgow Coma Scale; WBC, white blood cell; Hb, hemoglobin; Hct, hematocrit; Plt, platelet; PT, prothrombin time; APTT, activated partial thromboplastin time; T-Bil, total bilirubin; BUN, blood urea nitrogen; Cre, creatinine; CRP, C-reactive protein; APACHE II, Acute Physiology and Chronic Health Evaluation II; SOFA, Sequential Organ Failure Assessment. The data for the non-circulatory and circulatory failure groups are those at the time of admission to the ICU. For non-parametric continuous variables, medians and interquartile ranges (IQR) were presented, and categorical variables were expressed as numbers (*n*) and percentages (%). *p*-values were calculated by comparing the non-circulatory with the circulatory failure group using the Mann–Whitney U test and Fisher’s exact test. * *p* < 0.05, indicating a significant correlation.

**Table 2 diagnostics-14-02428-t002:** Multiple linear regression analysis of rSO_2_ in the knee in the circulatory failure group.

Outcome Variables	Independent Variables	B	SE	β	t	*p*	95% CI	VIF
rSO_2_ at the knee (0 h; Day 0)	Age	−0.08	0.07	−0.18	−1.13	0.27	−0.21–0.06	1.07
	Hb (0 h)	0.58	0.36	0.25	1.60	0.12	−0.16–1.32	1.03
	SOFA (day 0)	−0.56	0.28	−0.31	−1.98	0.06	−1.14–0.02	1.05
	R = 0.49, R^2^ = 0.24, Adjusted R^2^ = 0.17, F = 3.38, *p* = 0.03							
rSO_2_ at the knee (24 h; Day 1)	Age	−0.03	0.06	−0.08	−0.57	0.57	−0.15–0.09	1.01
	Hb (24 h)	0.61	0.45	0.20	1.36	0.19	−0.31–1.54	1.10
	SOFA (day 1)	−0.71	0.20	−0.53	−3.58	<0.01 *	−1.12–−0.3	1.11
	R = 0.64, R^2^ = 0.41, Adjusted R^2^ = 0.35, F = 6.72, *p* < 0.01 *							
rSO_2_ at the knee (48 h; Day 2)	Age	−0.17	0.08	−0.39	−2.04	0.06	−0.34–0.01	1.57
	Hb (48 h)	−0.03	0.60	−0.01	−0.06	0.96	−1.30–1.23	1.16
	SOFA (day 2)	−1.35	0.30	−0.87	−4.58	<0.01 *	−1.97–−0.74	1.53
	R = 0.74, R^2^ = 0.55, Adjusted R^2^ = 0.48, F = 7.76, *p* < 0.01 *							

Multiple linear regression analysis was performed to determine the factors influencing knee rSO_2_ in patients with circulatory failure. Multiple regression equations were created using age, Hb level, and SOFA score as independent variables that may influence rSO_2_. The results showed that SOFA score on Days 1 and 2 significantly influenced knee rSO_2_. Using the VIF, each independent factor was confirmed to be multicollinear. Using the Shapiro–Wilk test, the unstandardized residuals of these multiple regression models were confirmed to be normally distributed. * *p* < 0.05, indicating a significant correlation. SE, standard error; CI, confidence interval; rSO_2_, regional tissue oxygenation saturation; Hb, hemoglobin; SOFA, sequential organ failure assessment; VIF, variance inflation factor.

## Data Availability

The raw data supporting the conclusions of this article will be made available by the authors upon request.

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
