# Peer review of "Assessment of Microcirculatory Dysfunction by Measuring Subcutaneous Tissue Oxygen Saturation Using Near-Infrared Spectroscopy in Patients with Circulatory Failure"

_diagnostics, 2024, doi:10.3390/diagnostics14212428_

Round 1

Reviewer 1 Report

Comments and Suggestions for Authors

Material and methods:

·       Line 64: Please ensure the number of healthy subjects measured is reported in the results section. If no population size calculation was performed, this number should not be included in the materials and methods section.

·       The description of the three patient groups in the materials and methods section is unclear. To improve clarity, please provide a thorough description of each group, including inclusion and exclusion criteria, protocol details (such as probe location and the timing of measurements), and the recorded data (vital signs and any blood tests conducted). The goals are also not very clear.

·       The authors have not specified which vital signs are being recorded. While they mention GCS, HR, and systolic pressure (in table 1 in the results section), there is no explanation for the omission of diastolic pressure and MAP. Although providing a full clinical picture is important, it is unclear why these data are reported since they do not appear to be used in the subsequent analysis.

Results:

·       The exclusion of 48 patients due to age and 39 due to missing consent refers to individuals who still need to meet the inclusion criteria since consent is always required. Therefore, it's unclear why these exclusions are emphasized. Additionally, if the age threshold was set at 20 due to legal adulthood in Japan, those under 20 should not have been included in the study. However, if the study considers 18+ as full age, it could be interesting to compare this younger group against the typical 'ICU-age' population. It’s surprising that data from 48 individuals were collected yet not utilized. This aspect is reported for each group.

Was obtaining consent from the next of kin not an option, despite measurements being performed? This raises some confusion. Typically, the number of subjects who meet the inclusion criteria and provided consent (either themselves or via their next of kin) is reported. Following that, any instances where measurements do not meet other criteria, such as quality standards or other relevant factors, should be addressed to enhance the study's rigor. Can the authors please comment on this aspect?

·       Table 1 reports demographic and clinical data, not baseline? I do not see challenges performed in the protocol.

·       It is important to clarify whether patients in the ICU had multiple causes for admission. Specifically, for the 35 patients in the no circulatory failure group, the reported causes of admission total 40, indicating that some patients might have had more than one reason for ICU admission. Further details on the types of trauma or conditions leading to admission would provide insight into how these factors might influence the study results.

·       In line 164, ensure that a single value is provided for the median and the interquartile range (IQR) for clarity. For example, report the median and IQR as ‘Median = X, IQR = (Y, Z) or the 25th and 75th interquartile as reported in the table 1.

·       Verify if Figure 1 includes the rSO2 values reported for each location and group at admission. Clarify, whether all rSO2 measurements at different locations and for each group are presented in the figure, as this detail is crucial for understanding the data.

·       Include information on the number of days passed since ICU admission. This detail is important as it can significantly affect the results and interpretations, particularly in longitudinal assessments.

·       In the table reported in Figure 2, it is excessive to report so many decimal places for ρ and p values. For clarity, round these values consistently. The p-value is in lowercase; ensure standardized notation. This is true throughout the whole manuscript. For linear relationships, verify if Pearson’s correlation or another measure is more appropriate (in this case use R and not ρ). Report the units on the x-axis too. I believe the error bars are very small (if reported).

·       If the authors agree, restructuring the Results and Methods sections into subsections could enhance readability and clarity, justifying goals and results. This approach would differentiate the various components of the study and facilitate a more coherent presentation of the findings.

Discussion:

·       The authors suggest that subcutaneous rSO2 might be a better indicator than the mottling score. However, while the mottling score is reported, the manuscript does not provide results or evidence to support for this statement. To substantiate this claim, the authors should present comparative results or analysis demonstrating the relative efficacy of subcutaneous rSO2 versus the mottling score as indicators.

·       The manuscript suggests rSO2 may better predict severity. Including a Receiver Operating Characteristic (ROC) analysis would support this claim by demonstrating rSO2's predictive performance compared to other measures.

very minor:

·       Throughout the manuscript, reference numbers [Ref] incorrectly split sentences. Please review and correct this formatting to ensure they are placed at sentence or clause ends for clarity.

Author Response

Thank you for your kind and thorough reviews of our article. We have revised our manuscript according to the reviewers' suggestions. The revised sections are underlined. We would like to provide point-by-point responses to the reviewers' comments below. The modifications are as follows.

Answer to Comment:

Comment 1:

Material and methods:

  • Line 64: Please ensure the number of healthy subjects measured is reported in the results section. If no population size calculation was performed, this number should not be included in the materials and methods section.

Answer to Comment 1:

The number of volunteers has been moved to the Results section, as the population size was not calculated.

The changes are listed on page 4, line 163–165.

Comment 2:

  • The description of the three patient groups in the materials and methods section is unclear. To improve clarity, please provide a thorough description of each group, including inclusion and exclusion criteria, protocol details (such as probe location and the timing of measurements), and the recorded data (vital signs and any blood tests conducted). The goals are also not very clear.

Answer to Comment 2:

We have added details such as the selection and exclusion criteria and data recorded for the three patient groups.

The changes are listed on page 2, line 69–92 and page3, line 111–125.

Comment 3:

    The authors have not specified which vital signs are being recorded. While they mention GCS, HR, and systolic pressure (in table 1 in the results section), there is no explanation for the omission of diastolic pressure and MAP. Although providing a full clinical picture is important, it is unclear why these data are reported since they do not appear to be used in the subsequent analysis.

Answer to Comment 3:

The recorded vital signs have been added to the Methods section. The diastolic blood pressure and mean blood pressure have been added to Table 1. GCS and mean blood pressure have been reported because they are required when calculating the SOFA score. The systolic blood pressure has been reported because it is used as a criterion for circulatory failure.

 The changes are listed on page 3, line 111–124.

Comment 4:

Results:

  • The exclusion of 48 patients due to age and 39 due to missing consent refers to individuals who still need to meet the inclusion criteria since consent is always required. Therefore, it's unclear why these exclusions are emphasized. Additionally, if the age threshold was set at 20 due to legal adulthood in Japan, those under 20 should not have been included in the study. However, if the study considers 18+ as full age, it could be interesting to compare this younger group against the typical 'ICU-age' population. It’s surprising that data from 48 individuals were collected yet not utilized. This aspect is reported for each group. Was obtaining consent from the next of kin not an option, despite measurements being performed? This raises some confusion. Typically, the number of subjects who meet the inclusion criteria and provided consent (either themselves or via their next of kin) is reported. Following that, any instances where measurements do not meet other criteria, such as quality standards or other relevant factors, should be addressed to enhance the study's rigor. Can the authors please comment on this aspect?

Answer to Comment 4:

When the 2020 volunteer study began, the age of adulthood in Japan was 20 years; therefore, the study was started targeting the age group of 20 years and above. Although there were previous studies using Toccare on fetuses, there were no studies on children, and since it was not possible to recruit children as volunteers for this study either, the normal rSO2 values for children using Toccare were unknown. For this reason, both the non-circulatory and circulatory failure groups only included patients aged 20 years and above.

We mainly obtained consent from the patients themselves for both the non-circulatory and circulatory failure groups; however, when this was not possible, we obtained consent from the next of kin. Approximately, 48 people under the age of 20 years in the non-circulatory failure group fell under the exclusion criteria; therefore, consent forms were not obtained, and data was not collected. As a result, this data could not be used. Similarly, in the circulatory failure group, 129 cases of cardiac arrest met the exclusion criteria; therefore, consent forms were not obtained, and after excluding the other 92 cases from which consent could not be obtained, the final number of cases analyzed was 38. We have clarified the exclusion criteria in the main text.

The changes are listed on page 2, line 69–93 and page 4, line 163–173.

Comment 5:

       Table 1 reports demographic and clinical data, not baseline? I do not see challenges performed in the protocol.

Answer to Comment 5:

The titles in Table 1 have been rewritten to demographic and clinical data.

The changes are listed on page 4, line 174.

Comment 6:

 It is important to clarify whether patients in the ICU had multiple causes for admission. Specifically, for the 35 patients in the no circulatory failure group, the reported causes of admission total 40, indicating that some patients might have had more than one reason for ICU admission. Further details on the types of trauma or conditions leading to admission would provide insight into how these factors might influence the study results.
Answer to Comment 6:

The ICU at the Nihon University Hospital also accepts patients who are judged to be in a serious condition but who do not have any abnormalities in their vital signs; therefore non-circulatory failure group was also included in the study. The reasons for hospitalization in the non-circulatory failure group were trauma (n=15, 42.9%), cardiovascular disease (n=14, 40%), other internal medicine diseases (n=4, 11.4%), and overdose (n=2, 5.7%), for a total of 35 patients. The category of “trauma” included cervical spinal cord injuries and fractures of the extremities, while the category of “cardiovascular disease” included pleurisy and myocardial infarction.

The changes are listed on page 5, line 191–193.

Comment 7:

       In line 164, ensure that a single value is provided for the median and the interquartile range (IQR) for clarity. For example, report the median and IQR as ‘Median = X, IQR = (Y, Z) or the 25th and 75th interquartile as reported in the table 1.

Answer to Comment 7:

I have rewritten the data as median and interquartile range.

The changes are listed on page 7, line 215–222.

Comment 8:

       Verify if Figure 1 includes the rSO2 values reported for each location and group at admission. Clarify, whether all rSO2 measurements at different locations and for each group are presented in the figure, as this detail is crucial for understanding the data.

Answer to Comment 8:

I have also added the rSO2 in the circulatory failure group after admission to the ICU to Figure 1. All the rSO2 values measured are summarized in Figure 1.

 The changes are listed on page 6 and 7, line 202–207.

Comment 9:

       Include information on the number of days passed since ICU admission. This detail is important as it can significantly affect the results and interpretations, particularly in longitudinal assessments.

Answer to Comment 9:

Post-ICU admission, the changes in each parameter in the circulatory failure group have been added to the supplementary file Figure S1.

 The changes are listed on page 5, line 195–197.

Comment 10:

  • In the table reported in Figure 2, it is excessive to report so many decimal places for ρ and p values. For clarity, round these values consistently. The p-value is in lowercase; ensure standardized notation. This is true throughout the whole manuscript. For linear relationships, verify if Pearson’s correlation or another measure is more appropriate (in this case use R and not ρ). Report the units on the x-axis too. I believe the error bars are very small (if reported).

Answer to Comment 10:

In this study, we rounded off all the decimal places in the data. We changed the p-value to a lowercase. Since the data in this study is a non-parametric continuous variable, the Spearman correlation coefficient was used. Therefore, ρ was used as the correlation coefficient. The same content has been added to the Methods section. We have added the units for the x-axis in Figure 2.

 The changes are listed on page 8, line 229–230.

Comment 11:

  • If the authors agree, restructuring the Results and Methods sections into subsections could enhance readability and clarity, justifying goals and results. This approach would differentiate the various components of the study and facilitate a more coherent presentation of the findings.

Answer to Comment 11:

As suggested, we have added the subheadings.

Comment 12:

Discussion:

  • The authors suggest that subcutaneous rSO2 might be a better indicator than the mottling score. However, while the mottling score is reported, the manuscript does not provide results or evidence to support for this statement. To substantiate this claim, the authors should present comparative results or analysis demonstrating the relative efficacy of subcutaneous rSO2 versus the mottling score as indicators.

Answer to Comment 12:

As highlighted, we were unable to find any results showing the effectiveness of subcutaneous rSO2 by directly comparing it with the mottling score. For this reason, we decided to exclude the mottling score from this study.

Comment 13:

       The manuscript suggests rSO2 may better predict severity. Including a Receiver Operating Characteristic (ROC) analysis would support this claim by demonstrating rSO2's predictive performance compared to other measures.

Answer to Comment 13:

As highlighted, we have have added the ROC curve to Figure 4.

 The changes are listed on page 10, line 264–275.

Comment 14:

very minor:

  • Throughout the manuscript, reference numbers [Ref] incorrectly split sentences. Please review and correct this formatting to ensure they are placed at sentence or clause ends for clarity.

Answer to Comment 14:

Reference numbers have been moved to the end of the sentence or clause as citations.

Reviewer 2 Report

Comments and Suggestions for Authors

The study provides useful insight into the use of subcutaneous tissue oxygen saturation as a non-invasive, real-time marker for microcirculatory dysfunction and organ damage in patients with circulatory failure. The study looks convincing, and the comprehensive statistical analysis leaves a pleasant impression.

However, there are some issues that need to be addressed.

1. The main goal of the authors is to study microcirculation dysfunction in patients with circulatory failure. At the same time, the authors chose an indirect method for assessing microcirculation, when a number of other direct methods of measuring blood flow could be used, for example, laser Doppler flowmetry [see e.g. doi.org/10.3389/fphys.2019.00416] or laser speckle contrast imaging [doi.org/10.1109/TBME.2019.2950323]. At least discussions about this should be added to the introduction.

2. Also, the study relies on rSO2 measurements as the main investigative tool without comparing its effectiveness to other more established methods, such as invasive blood gas monitoring, Doppler ultrasound or the above-mentioned optical techniques.

3. NIRS is sensitive to variables such as skin pigmentation, tissue thickness, ambient light, etc., which were not controlled during the study. These factors may affect the accuracy of rSO2 measurements. Please discuss this.

4. Why did the authors choose these particular measurement zones?

5. I think it is more appropriate to move the first paragraph of the section “3. Results” to the section “Study population”.

6. Why did the authors exclude patients younger than 20 years old? The results of the study are not applicable to younger groups of the population, who could benefit from such an approach to monitoring in pediatric ICUs.

7. In Figure 2, the table looks out of place. Please arrange it in the form of a separate table.

8. Please check Figure S1 in the Supplementary file (hieroglyphs in the designation of the ordinate axis).

In general, future studies involving larger and more diverse patient groups are needed to confirm the obtained results and explore the practical application of rSO2 monitoring in clinical conditions.

Author Response

Thank you for your kind and thorough reviews of our article. We have revised our manuscript according to the reviewers' suggestions. The revised sections are underlined. We would like to provide point-by-point responses to the reviewers' comments below. The modifications are as follows.

Answer to Comment:

Comment 1:

Comments and Suggestions for Authors

The study provides useful insight into the use of subcutaneous tissue oxygen saturation as a non-invasive, real-time marker for microcirculatory dysfunction and organ damage in patients with circulatory failure. The study looks convincing, and the comprehensive statistical analysis leaves a pleasant impression.

However, there are some issues that need to be addressed.

  1. The main goal of the authors is to study microcirculation dysfunction in patients with circulatory failure. At the same time, the authors chose an indirect method for assessing microcirculation, when a number of other direct methods of measuring blood flow could be used, for example, laser Doppler flowmetry [see e.g. doi.org/10.3389/fphys.2019.00416] or laser speckle contrast imaging [doi.org/10.1109/TBME.2019.2950323]. At least discussions about this should be added to the introduction.

Answer to Comment 1

We have added an explanation in the introduction as to why we chose subcutaneous rSO2 for microcirculation assessment.

 The changes are listed from line 37 on page 1 to line 52 on page 2.

Comment 2:

  1. Also, the study relies on rSO2 measurements as the main investigative tool without comparing its effectiveness to other more established methods, such as invasive blood gas monitoring, Doppler ultrasound or the above-mentioned optical techniques.

Answer to Comment 2

As this study did not compare rSO2 measurement with other methods of evaluating microcirculation, it was not possible to determine which was superior. We have added this point to the limitations section.
 The changes are listed on page 12, line 350–353.

Comment 3:

  1. NIRS is sensitive to variables such as skin pigmentation, tissue thickness, ambient light, etc., which were not controlled during the study. These factors may affect the accuracy of rSO2 measurements. Please discuss this.

Answer to Comment 3

As highlighted, this study has not controlled for errors in NIRS caused by skin pigmentation, tissue thickness, or ambient light. We have added this point to the limitations section.

The changes are listed on page 12, line 347–348.

Comment 4:

  1. Why did the authors choose these particular measurement zones?

Answer to Comment 4:

We have added a description of how to select the measurement site for Toccare in the Methods section.

The changes are listed on page 3, line 98–104.

Comment 5:

  1. I think it is more appropriate to move the first paragraph of the section “3. Results” to the section “Study population”.

Answer to Comment 5:

We have moved the first paragraph of the Results to the section “Study population”.

 The changes are listed on page 2, line 74–84.

Comment 6:

  1. Why did the authors exclude patients younger than 20 years old? The results of the study are not applicable to younger groups of the population, who could benefit from such an approach to monitoring in pediatric ICUs.

Answer to Comment 6:

Toccare is a new device, and there are no studies of its use in children. As we were unable to recruit any pediatric participants for this volunteer study, the normal range of rSO2 measured by Toccare in children remains unknown. For this reason, we also excluded patients under 20 years of age from the non-circulatory and circulatory failure groups.

 The changes are listed on page 2, line 90–93.

Comment 7:

  1. In Figure 2, the table looks out of place. Please arrange it in the form of a separate table.

Answer to Comment 7:

We have changed the table in Figure 2 to the appropriate position.

 The changes are listed on page 8, line 229–230.

Comment 8:

  1. Pleasecheck Figure S1 in the Supplementary file (hieroglyphs in the designation of the ordinate axis).

Answer to Comment 8:

The original supplementary file Figure 1 has been changed to Figure 1 in the main text.

The changes are listed on page 6.

Comment 9:

In general, future studies involving larger and more diverse patient groups are needed to confirm the obtained results and explore the practical application of rSO2 monitoring in clinical conditions.

Answer to Comment 9

We have added a limitation that research is needed on a larger and more diverse patient groups.

The changes are listed on page 12, line 349–350.

Round 2

Reviewer 2 Report

Comments and Suggestions for Authors

The authors responded to the comments. This manuscript can be accepted.